# Decline of Anti-SARS-CoV-2 IgG Antibody Levels 6 Months after Complete BNT162b2 Vaccination in Healthcare Workers to Levels Observed Following the First Vaccine Dose

**DOI:** 10.3390/vaccines10020153

**Published:** 2022-01-20

**Authors:** Oktavija Đaković Rode, Kristian Bodulić, Sanja Zember, Nataša Cetinić Balent, Anđa Novokmet, Marija Čulo, Željka Rašić, Radojka Mikulić, Alemka Markotić

**Affiliations:** 1Department for Clinical Microbiology, Division for Virology, University Hospital for Infectious Diseases “Dr. Fran Mihaljević”, 10000 Zagreb, Croatia; ncetinic@bfm.hr (N.C.B.); rmikulic@bfm.hr (R.M.); 2School of Dental Medicine, University of Zagreb, 10000 Zagreb, Croatia; 3Research Department, University Hospital for Infectious Diseases “Dr. Fran Mihaljević”, 10000 Zagreb, Croatia; kbodulic@bfm.hr (K.B.); amarkotic@bfm.hr (A.M.); 4Health Care Quality Unit, University Hospital for Infectious Diseases “Dr. Fran Mihaljević”, 10000 Zagreb, Croatia; szember@bfm.hr (S.Z.); zrasic@bfm.hr (Ž.R.); 5Department for Adult Intensive Care and Neuroinfections, University Hospital for Infectious Diseases “Dr. Fran Mihaljević”, 10000 Zagreb, Croatia; anovokmet@bfm.hr; 6Department for Clinical Microbiology, Division for Bacteriology, Hospital Infections and Sterilization, University Hospital for Infectious Diseases “Dr. Fran Mihaljević”, 10000 Zagreb, Croatia; mculo@bfm.hr; 7School of Medicine, Catholic University of Croatia, 10000 Zagreb, Croatia; 8Faculty of Medicine, University of Rijeka, 51000 Rijeka, Croatia

**Keywords:** 6 month longitudinal study, anti-SARS-CoV-2 antibodies, BNT162b2 vaccine, healthcare workers, Croatia

## Abstract

Research on post-vaccination antibody dynamics has become pivotal in estimating COVID-19 vaccine efficacy. We studied anti-SARS-CoV-2 Spike RBD IgG levels in 587 healthcare workers (2038 sera) who completed BNT162b2 vaccination. Average antibody titer 3 weeks after the first dose in COVID-19-naïve participants (median 873.5 AU/mL) was 18-fold higher than the test threshold, with a significant increase 1 month (median 9927.2 AU/mL) and an exponential decrease 3 (median 2976.7 AU/mL) and 6 (median 966.0 AU/mL) months after complete vaccination. Participants with a history of COVID-19 prior to vaccination showed significantly higher antibody levels, particularly after the first dose (median 14,280.2 AU/mL), with a slight decline 1 month (median 12,700.0 AU/mL) and an exponential decline in antibody titers 3 (median 4831.0 AU/mL) and 6 (median 1465.2 AU/mL) months after vaccination. Antibody levels of COVID-19-naïve subjects after the first dose were moderately correlated with age (*r* = −0.4). Multivariate analysis showed a strong independent correlation between IgG levels 6 months after vaccination and both IgG titers after the first dose and 1 month after vaccination (*R*^2^ = 0.709). Regardless of pre-vaccination COVID-19 history, IgG levels 6 months after vaccination were comparable to antibody levels reached by COVID-19-naïve participants after the first vaccine dose.

## 1. Introduction

The pandemic of severe acute respiratory coronavirus type 2 (SARS-CoV-2) has made us witnesses of historical events. In 2003, SARS-CoV-1 emerged and displayed a strong pathogenic potential that had not been recorded among coronaviruses. Fortunately, SARS-CoV-1 soon disappeared, with its current existence assumed in unknown hosts. On the other hand, SARS-CoV-2 has adapted to humans much more efficiently, mutating and creating new variants that successfully evade the immune response. While modern molecular technology has enabled fast vaccine development, assessing the immune response to an emerging pathogen requires time. Longitudinal infection monitoring shows great promise in the analysis of antibody kinetics, whose results can be implemented in infection prevention [1,2,3,4]. 

Following SARS-CoV-2 infection, immunocompetent patients rapidly develop specific antibodies, whose levels generally show a positive correlation with disease severity [3,4,5,6,7]. Relatively high antibody levels in severe COVID-19 raise the question of their efficiency in immune defense. Furthermore, recent studies showed a high agreement between antibody neutralization ability and anti-SARS-CoV-2 IgG levels [8,9,10,11]. Assuming that levels of anti-SARS-CoV-2 IgG correlate with the duration of effective immunity, antibody kinetics monitoring could prove useful as a possible predictor for revaccination planning. While neutralizing antibodies could play an important role in the clinical course of COVID-19, cellular immunity appears to be even more significant [2,7,12,13,14].

The analysis of specific antibodies in patients with a history of COVID-19 linked the immune response to various factors, including disease stage and severity, age, and comorbidities. While IgG antibodies can be detected in a vast majority of COVID-19 patients 10 days after the infection, their levels generally decrease over time [4,5,6,7]. Similar patterns in antibody dynamics could be predicted after vaccination [15,16,17,18,19,20,21]. Post-COVID-19 specific antibodies are directed toward different antigens, such as the nucleoprotein (NP) and spike (S) antigens. Considering that mRNA vaccines are based on the S1 subunit of the recombinant S antigen, post-vaccination assessment of immunity is only feasible with S1-specific serological tests [19,21,22,23,24,25].

In this longitudinal prospective study, we analyzed the IgG antibody response after vaccination with the BNT162b2 SARS-CoV-2 mRNA vaccine in healthcare workers (HCWs). The humoral immunity and COVID-19 status were evaluated at four timepoints: 3 weeks after the first dose, and 1, 3, and 6 months after complete vaccination. Data were analyzed according to participants’ age, sex, workplace, and pre-vaccination COVID-19 history. This study aimed to assess the post-vaccination anti-SARS-CoV-2 IgG dynamics, which could prove useful in predicting the need for revaccination according to antibody levels at different timepoints after vaccination.

## 2. Materials and Methods

Vaccination of HCWs with the BNT162b2 mRNA SARS-CoV-2 vaccine started on 27 December 2020 at the University Hospital for Infectious Diseases, which is a tertiary care teaching hospital in Zagreb and the leading institution for the treatment of COVID-19 in Croatia. A two-dose regimen of the PfizerBioNTech COVID-19 vaccine BNT162b2 was administered in all HCWs, 3 weeks apart. Prior to receiving the second vaccine dose, all HCWs were offered to participate in a longitudinal prospective study to monitor the immune response after vaccination. All voluntarily included HCWs signed the informed consent. Data were collected on participants’ demographics, workplace, and a history of COVID-19 at different timepoints. The study was approved by the Institutional Clinical Research Ethics Committee. 

Consecutive blood samples were collected at four timepoints: 3 weeks after the first BNT162b2 vaccine dose, and then 1, 3, and 6 months after the second vaccine dose from January 2021 to September 2021. All samples were stored at the temperature of −20 °C until testing. Anti-SARS-CoV-2 antibody levels were measured using a quantitative chemiluminescent microparticle immunoassay for antibodies against the receptor-binding domain (RBD) of S1 subunit of the SARS-CoV-2 spike protein (CMIA, SARS-CoV-2 IgG II Quant, Architect, Abbott). According to the manufacturer’s instructions, a threshold of 50 AU/mL was used to identify anti-SARS-CoV-2 IgG-positive immune response after vaccination. The analytical measuring interval was 21–80,000 AU/mL. The manufacturer declared 99.4% and 99.6% positive and negative predictive agreement, respectively, with 100% (95% CI: 95.72–100%) positive agreement with neutralization testing results [11,21,25]. 

Statistical analysis and data visualization were done in R (version 4.1.0.) with ggpubr (version 0.4.0.) and ggplot2 (version 3.3.5.) packages [26]. Independent nonparametric distributions were compared with Mann–Whitney U test in the case of two-group comparisons and Kruskal–Wallis test in the case of multiple-group comparisons. Paired sample distributions were compared using Wilcoxon signed-rank test. Pairwise correlations between numerical variables were analyzed using the correlation test and Spearman’s rank coefficient. All tests were two-sided with a significance level of 95%. The *p*-values were adjusted for multiple comparisons with the Bonferroni method. Multiple linear regression was used to evaluate the relationship between IgG titers at different timepoints and predictors of interest, with log transformation applied to IgG titer values and age. Variable selection was performed with the best subset selection method. Residual distribution normality was assessed by residual versus fit plots and quantile–quantile plots.

## 3. Results

### 3.1. Study Participants

A total of 587 HCWs vaccinated with two BNT162b2 vaccine doses administered 3 weeks apart were recruited for study participation, among whom 417 (71.04%) worked in prolonged close contact with COVID-19 patients. Anti-SARS-CoV-2 IgG antibody levels were determined in 2038 serum samples, which included 575 samples obtained 3 weeks after the first dose (S1), 530 samples obtained 1 month after the second dose (S2), 499 samples obtained 3 months after the second dose (S3), and 434 samples obtained 6 months after the second vaccine dose (S4). The mean age of participants was 42.3 years (SD 12.3, range 19.1–66.5), and 482 (82.1%) of the participants were female. Due to close contact with a COVID-19 patient outside of workplace, 121 (20.8%) HCWs reported self-isolation before vaccination and tested negative for SARS-CoV-2. Additionally, COVID-19 diagnosis before vaccination was confirmed in 86 (14.7%) participants, 74 of whom had had a positive PCR test result before vaccination. Diagnosis of pre-vaccination COVID-19 for the other 12 participants was retrospectively defined according to a positive anti-NP serological test at S1. Thirteen (2.21%) participants had a breakthrough SARS-CoV-2 infection after vaccination, among whom one HCW had a positive test result between S1 and S2, seven HCWs had a positive test result between S2 and S3, and five HCWs had a positive test result between S3 and S4. Sera of these HCWs obtained after COVID-19 onset were not included in the antibody dynamics analysis or in the group comparisons or regression analyses. 

### 3.2. Anti-SARS-CoV-2 IgG Antibody Dynamics

We monitored the humoral immune response to SARS-CoV-2 mRNA vaccine in HCWs at different periods. The distribution of antibody levels at defined timepoints stratified by COVID-19 history is shown in Figure 1. Participants without a history of COVID-19 showed an increase in IgG levels at S2 (median 9927.2 AU/mL) when compared to S1 (median 873.5 AU/mL, *p* < 0.001). Likewise, we recorded a relatively high ratio of participants without COVID-19 history with IgG levels between 2000 and 10,000 AU/mL (45.4%), as well as between 10,000 and 20,000 AU/mL (30.7%) at S2. Antibody levels at S3 (median 2976.7 AU/mL) were significantly lower than antibody levels at S2 in COVID-19-negative participants (*p* < 0.001). Additionally, these participants exhibited significantly lower IgG levels at S4 (median 966.0 AU/mL) when compared to antibody levels at S3 (*p* < 0.001). When contrasting antibody levels of participants without a COVID-19 history at S4 and S1, no significant difference was found (*p* = 0.300). Moreover, the antibody titer range of 50–2000 AU/mL was the most represented titer range at S1 (77.7% of participants) and S4 (85.3% of participants). Seven (1.2%) HCWs without a COVID-19 history had undetectable IgG antibodies at S1. Two of them received immunosuppressive therapy and did not develop antibodies at S2, while also remaining seronegative at S3 and S4.

When analyzing antibody levels of participants who had recovered from COVID-19 before vaccination, no significant differences were found between S1 (median 14,280.2 AU/mL) and S2 (median 12,700.0 AU/mL, *p* = 0.999). On the other hand, participants with a history of COVID-19 showed a significant decrease in antibody levels at S3 (median 4831.0 AU/mL) when compared to antibody titer at S2 (*p* < 0.001). Similarly, HCWs with a history of COVID-19 exhibited a significant decrease in antibody levels at S4 (median 1465.2 AU/mL) when compared to antibody levels at S3 (*p* < 0.001). When assessing the relationship between antibody levels of HCWs who had recovered from COVID-19 and the disease onset time relative to vaccination, no significant correlations were found at any of the analyzed timepoints (S1: *r* = 0.27, S2: *r* = 0.08, S3: *r* = 0.24, S4: *r* = 0.22, *p* > 0.050). Furthermore, antibody levels of participants with a history of COVID-19 were significantly higher than antibody levels of COVID-19 negative participants at S1, S3, and S4 when adjusting for multiple comparisons (Table 1).

We also analyzed anti-SARS-CoV-2 IgG titer ratios between consecutive samples (Figure 2). We found that HCWs with a pre-vaccination history of COVID-19 showed lower antibody level ratios between S2 and S1 than COVID-19-naïve HCWs (medians 0.96 and 11.18, *p* < 0.001). Interestingly, these ratios did not depend on the time of COVID-19 onset relative to vaccination (*r* = −0.03, *p* = 0.817). Out of 72 HCWs with a history of COVID-19, 40 showed a decrease in antibody levels between S2 and S1 (missing S1 or S2 data for 14 participants). Furthermore, median antibody ratio between S3 and S2 was 0.39 in HCWs who had contracted COVID-19 before vaccination and 0.30 in COVID-19-naïve HCWs. Again, this difference was statistically significant (*p* < 0.001). Median antibody ratio between S4 and S3 was also higher in HCWs with a history of COVID-19 when compared to COVID-19-naïve HCWs (medians 0.39 and 0.35, *p* = 0.009). Finally, eight and six participants showed an increase in antibody levels in the S2–S3 and S3–S4 timespans, respectively. All of these participants had COVID-19 after vaccination.

### 3.3. Anti-SARS-CoV-2 IgG Antibody Levels by Age, Sex, and COVID-19 Patient Contact

Next, we analyzed anti-SARS-CoV-2 IgG levels by age group, sex, and workplace of participants in close contact with COVID-19 patients. The results of these comparisons are outlined in Table 1. We found a significant difference between antibody titers in different age groups, where participants in the 18–30 years group showed the highest antibody titers at all timepoints and participants in the >60 years group showed the lowest antibody titers at S1, S2, and S4. These differences were most extreme at S1 and decreased at later timepoints. We also evaluated the effect of pre-vaccination COVID-19 status on the correlation between age and antibody levels. While we did find a moderate negative correlation between age and antibody titer 3 weeks after the first vaccine dose in COVID-19-naïve participants (*r* = −0.4, *p* < 0.001), no correlation of such type was found in participants with a history of COVID-19 (*r* = 0.17, *p* = 0.104). At later timepoints, the strength of this correlation lowered for HCWs without a history of COVID-19 (S2: r = −0.24, S3: *r* = −0.25, S4: *r* = −0.23, *p* < 0.001) and remained insignificant for HCWs who had recovered from COVID-19 before vaccination. Participants with a history of COVID-19 showed higher antibody titers at S1, S3, and S4, regardless of age group. Further analysis of S2 IgG levels showed that COVID-19-naïve HCWs had significantly higher antibody titers in the 18–30 age group (medians 14,111 AU/mL vs. 9295 AU/mL, *p* = 0.007), but significantly lower IgG levels than COVID-19 recovered HCWs in older age groups (41–50 age group: medians 7924.5 AU/mL vs. 11,263.35 AU/mL, *p* = 0.012, 51–60 age group: medians 7053.9 AU/mL vs. 12,483 AU/mL, *p* = 0.014). No significant differences were found between COVID-19-naïve and HCWs with a history of COVID-19 in the 31–40 age group (medians 10,280.0 AU/mL vs. 11,099 AU/mL, *p* = 0.516). When comparing antibody titers between sexes, we did not find a statistically significant difference between male and female participants at any of the analyzed timepoints. In addition, we hypothesized that the transmission of a small concentration of SARS-CoV-2, which is insufficient to cause COVID-19, could lead to higher IgG titers in HCWs in prolonged close contact with COVID-19 patients. These HCWs reached significantly higher antibody titers than other HCWs at S1, S3, and S4.

### 3.4. Multiple Regression Models of Anti-SARS-CoV-2 IgG Antibody Levels

We used multiple linear regression to explore the relationship between potentially important predictors and IgG titer at S1 in a multivariate setting (Table 1). These predictors included subjects’ age, sex, contact with COVID-19 patients, and history of COVID-19 before vaccination. When eliminating the effects of other predictors, we found a significant negative correlation between the subjects’ age and IgG titer at S1 (an average of 1.28% decrease in antibody titer per 1% increase in age, *p* < 0.001). Similarly, we found a significant positive correlation between a history of COVID-19 before vaccination and IgG titer at S1 (an average of 817.6% increase in antibody titer in case of pre-vaccination history of COVID-19, *p* < 0.001). We also used multiple linear regression to assess the relationship between antibody titer at S4 and antibody titers measured at S2 and S1, along with the already stated predictors. When removing the effects of other predictors, we found a significant positive correlation between antibody levels at S4 and antibody levels both at S2 (an average of 0.63% increase in S4 antibody titer per 1% increase in S2 antibody titer, *p* < 0.001) and S1 (an average of 0.15% increase in S4 antibody titer per 1% increase in S1 antibody titer, *p* < 0.001). Moreover, the stated combination of IgG titers explained 70.9% of the variation in the IgG titer at S4.

## 4. Discussion

Our prospective longitudinal study contributes to the finding of vastly dynamic humoral immunity induced by BNT162b2 vaccination during a 6 month follow-up. Significant differences in anti-SARS-CoV-2 antibody kinetics were observed between COVID-19-naïve subjects and subjects who had recovered from COVID-19 prior to vaccination. However, regardless of participants’ pre-vaccination COVID-19 history, IgG levels 6 months after complete vaccination were comparable to antibody levels reached by COVID-19-naïve participants after the first vaccine dose.

It has been confirmed that the anti-SARS-CoV-2 S1 RBD assay used in our study shows a strong correlation to antibody neutralization ability, allowing us to estimate the immunogenic potential of the BNT162b2 mRNA vaccine in different participant groups [11,19,21,25]. Antibody levels in non-COVID-19 participants reached as early as 3 weeks after the first dose were on average 18-fold higher than the test threshold, highlighting the vast immunogenic potential of the BNT162b2 vaccine. Furthermore, 1 month after the second vaccine dose, we registered an 11-fold antibody level increase in COVID-19-naïve participants, which emphasizes the importance of the second vaccine dose administration. Subjects with a history of COVID-19 developed a specific pattern of immune response after vaccination. These HCWs exhibited significantly higher IgG levels after the first vaccine dose. This could be attributed to specific memory cells induced by the disease, confirming that post-COVID-19 memory B-cells recognize the S antigen in the vaccine. However, this difference was much less apparent 1 month after the second vaccine dose. Stated results suggest that participants with a history of COVID-19 before vaccination reached the plateau in antibody titer after the first vaccine dose, with most of them showing a decrease or a relatively small increase in antibody levels after the second dose. Accordingly, the second dose in the vast majority of subjects who had COVID-19 prior to vaccination does not appear to have a significant effect in achieving a higher plateau of humoral immune response. One dose of vaccine received after COVID-19 is sufficient to produce peak antibody levels extremely quickly. These findings agree with the results of previous studies and go in line with the recommendation of second dose administration 6 months after recovering from COVID-19 [14,15,16,17,18,19,20,21,22,23,24,25,27,28,29,30]. Furthermore, these results can be explained by the limited humoral response to SARS-CoV-2 and the self-maintenance nature of the immune system. Additionally, changes in antibody levels 1 month after the second dose in HCWs with a history of COVID-19 were not correlated to the time of disease onset or age and likely depended on the clinical presentation of the disease [23,28,29]. 

When analyzing antibody kinetics 3 and 6 months after the second vaccine dose, we recorded an increase in antibody levels in HCWs with a history of COVID-19 after the second vaccine dose. The stated increase most likely resulted from a humoral immune response to SARS-CoV-2. Antibody levels in other participants showed an exponential decline 3 and 6 months after the second dose, where levels in HCWs with a history of COVID-19 before vaccination decreased more moderately. Both higher antibody plateau levels and this slower decline resulted in higher antibody titers 3 and 6 months after the second vaccine dose in HCWs with a pre-vaccination history of COVID-19. However, antibody levels 6 months after the second dose decreased significantly when compared to plateau values regardless of participants’ history of COVID-19. Stated titers were similar to antibody titer of COVID-19-naïve HCWs 3 weeks after the first vaccine dose. These results go in favor of a third dose administration 6 months after vaccination regardless of the pre-vaccination COVID-19 status. However, further studies on the long-term functionality of immunity are needed, particularly regarding the differences between individuals with a different pre-vaccination COVID-19 status. Additionally, our multiple regression model demonstrated an independent correlation between antibody titer 6 months after the second dose and both antibody titers 3 weeks after the first dose and 1 month after the second vaccine dose. This suggests that the humoral response after the first vaccine dose and the peak immune response after the second dose may prove useful in predicting antibody levels 6 months after complete vaccination and potentially at future timepoints. This represents a novel finding not discussed in the related literature so far. 

The finding of moderate negative correlation between antibody levels and age also agrees with the results of previous relevant studies [6,16]. This is especially true regarding the relatively high antibody levels in the youngest age groups and relatively low antibody levels in HCWs older than 60 years. This finding could be explained by a slower immune response characteristic for the elderly. Nevertheless, the correlation between age and antibody titer lowered significantly after the second vaccine dose. Furthermore, our multivariate analysis showed no direct correlation between age and antibody titer 6 months after the second dose. Accordingly, participants’ age did not affect the antibody plateau values or antibody dynamics after the second vaccine dose, further supporting the argument of a strong immunogenic potential of the vaccine. However, it should be noted that this analysis did not include participants older than 66 years. We expect that these participants would show even lower antibody levels after the first vaccine dose and possibly lower antibody titer after the second dose. When considering sex, the higher medians of antibody titers in female participants at all timepoints are consistent with the results of similar studies [6,22,23,31]. However, these differences were not statistically significant, which was further confirmed by our multiple regression analysis. It should be noted that the distributions of participants with a history of COVID-19 were extremely similar across age and sex groups. Consequently, the history of COVID-19 did not significantly impact the results of these comparisons, as confirmed by multiple regression models. 

It is well known that SARS-CoV-2 is an enveloped droplet-borne virus sensitive to disinfectants. Mandatory use of standard precautions that include hand hygiene and face protection can successfully prevent SARS-CoV-2 infection. Considering that the hospital setting can hold great risk for SARS-CoV-2 infection, antibody level follow-up could prove important in identifying asymptomatic infections and natural boosters in HCWs [19,25,27,32]. Even though bivariate analyses showed significantly higher antibody levels in HCWs in prolonged close contact with COVID-19 patients, this was disproved by multivariate regression models. Age seemed to be a confounding variable in the bivariate analysis, with participants in prolonged contact with COVID-19 patients being significantly younger. Considering that the vast majority of HCWs who had COVID-19 were infected with SARS-CoV-2 outside of their workplace, these findings suggest a proper use and high effectiveness of protective measures in our hospital, which was already discussed in a recent study [33].

The main limitation of this study is the lack of data on antibody levels prior to vaccination, which did not allow for a detailed assessment of the antibody dynamics, especially regarding the participants who had recovered from COVID-19 before vaccination. Another shortfall of this study is the potential misclassification of participants according to the history of COVID-19 before vaccination. While most of the analyzed HCWs with confirmed COVID-19 were tested by PCR, a smaller number of HCWs had their COVID-19 diagnosis established retrospectively with NP serological tests. Since BNT162b2 vaccine is based on the recombinant S antigen, NP-specific antibodies are of great help in the retrospective diagnosis of COVID-19. However, a common pitfall of this approach is a relatively quick decay of NP-specific antibodies after disease onset [2,3,7,8,9,10,21,24,25,34]. Although we expect that the vast majority of participants with a history of COVID-19 were correctly identified, asymptomatic cases may have been missed. However, we believe that this did not significantly change the results and the conclusions of this study. Lastly, our test subjects represent a convenient sample that consisted of our hospital’s HCWs, with a lack of participants younger than 18 and older than 66 years. Consequently, the results of this study should only be generalized to the working-age population.

## 5. Conclusions

After completing two-dose BNT162b2 vaccination, HCWs showed a decrease in anti-SARS-CoV-2 IgG antibody levels as early as 3 months. History of COVID-19 showed a significant influence on antibody kinetics in vaccinated subjects. Unlike at the other three timepoints, where subjects with a history of COVID-19 always seemed to show a higher level of antibodies, 1 month after the second dose, the antibody levels in subjects with a history of COVID-19 were similar to those in COVID-19-naïve participants. The COVID-19-naïve participants exhibited a multifold increase in antibody levels 1 month after the second dose which differs from kinetics in participants with a history of COVID-19. However, 6 months after vaccination, regardless of pre-vaccination history of COVID-19, all participants had anti-SARS-CoV-2 IgG levels similar to antibody levels reached by COVID-19-naïve participants 3 weeks after the first dose of vaccine. Our results suggest a strong decline in IgG levels 6 months after vaccination with two doses, which was strongly correlated with IgG levels 3 weeks after the first dose, as well as with IgG levels achieved 1 month after the second dose. However, further monitoring is needed to elucidate the relationship between IgG decay and infection protection.

## Figures and Tables

**Figure 1 vaccines-10-00153-f001:**
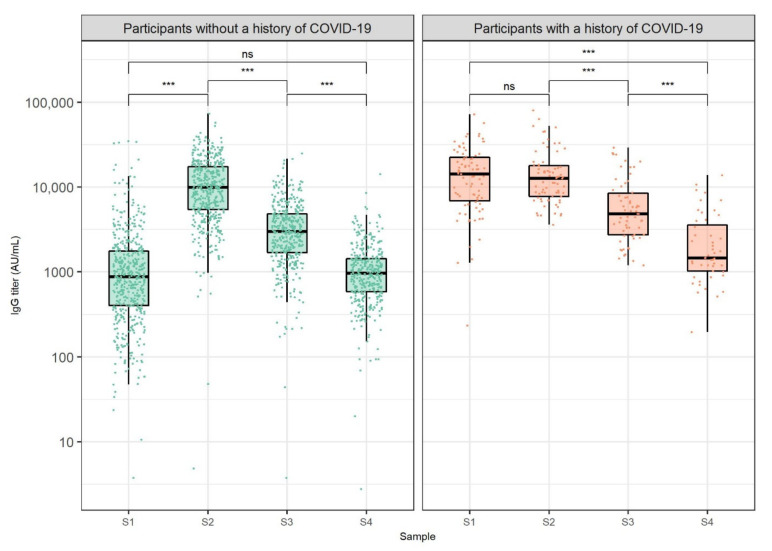
Anti-SARS-CoV-2 IgG titers 3 weeks after the first dose (S1) and 1 (S2), 3 (S3), and 6 months (S4) after the second dose of BNT162b2 vaccine against COVID-19, stratified by pre-vaccination COVID-19 status. The boxes show the median and interquartile range of the distribution, while the whiskers extend to the minimum and maximum nonoutlier values of the distribution. Points denote individual participants. The *y*-axis is logarithmically scaled. Sera of participants obtained after COVID-19 onset were not included. ns: non-significant, *** *p* < 0.001 (Wilcoxon signed-rank test, *p*-values adjusted for multiple comparisons with the Bonferroni method).

**Figure 2 vaccines-10-00153-f002:**
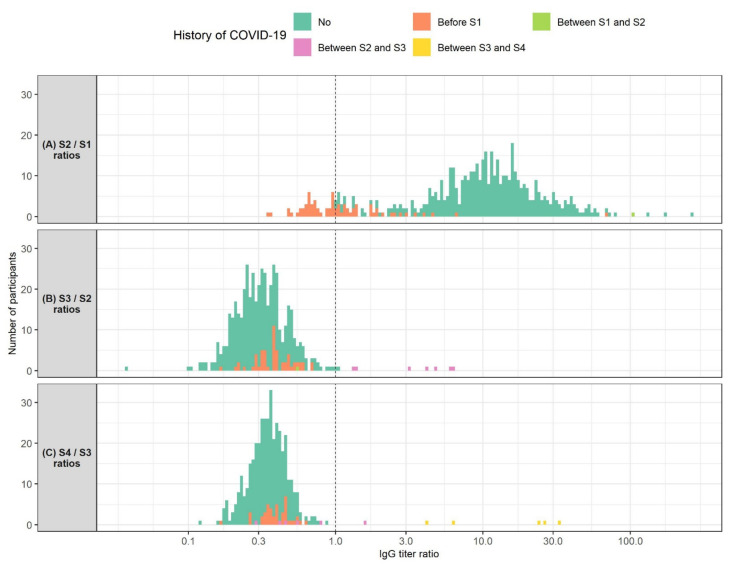
Histogram of ratios of SARS-CoV-2 IgG levels measured: (A) 1 month after the second BNT162b2 vaccine dose (S2) and 3 weeks after the first vaccine dose (S1) (*N* = 524); (B) 3 months after the second vaccine dose (S3) and 1 month after the second vaccine dose (*N* = 479); (C) 6 months after the second vaccine dose (S4) and 3 months after the second vaccine dose (*N* = 406). Participants are colored according to their history of COVID-19. The *X*-axis is logarithmically scaled for increased resolution. The dotted line marks an IgG titer ratio of 1 which corresponds to no changes in IgG titer between the respective timepoints. Participants shown left of the dotted line exhibited a decrease in IgG titers between the respective timepoints. Participants shown right of the dotted line showed an increase in IgG titers between the respective timepoints.

**Table 1 vaccines-10-00153-t001:** Bivariate and multivariate analyses of anti-SARS-CoV-2 IgG antibody titers in healthcare workers after vaccination with BNT162b2 mRNA vaccine.

Post- Vaccination IgG Titer	Anti-SARS-CoV-2 IgG Titer (AU/mL Median (IQR))											
	Participants (*n* = 587)	Pre-Vaccination COVID-19 History		Age (years)					Sex		Workplace with Patient Contacts	
		Yes (*n* = 86)	No (*n* = 501)	18–30 (*n* = 143)	31–40 (*n* = 96)	41–50 (*n* = 177)	51–60 (*n* = 142)	>60 (*n* = 29)	Male (*n* = 105)	Female (*n* = 482)	Yes (*n* = 417)	No (*n* = 170)
S1 (*N* = 575)	1050.8 (460.4–3138.5)	14,280.2 (6913.4–22347.7)	873.5 (402.9–1753.3)	1951.9 (1064.9–4775.6)	1140.4 (565.0–3484.6)	851.2 (289.8–1986.3)	751.7 (298.5–1654.8)	460.2 (151.1–1161.0)	931.6 (350.0–2274.8)	1071.4 (494.0–3155.4)	1140.4 (565.0–3484.6)	851.2 (289.8–1986.3)
*p*-value		<0.001		<0.001					0.087		<0.001	
*p*-adj ^#^		<0.001		<0.001					0.990		0.011	
S2 ^*^ (*N* = 529)	10,238.6 (5736.4–17,693.0)	12,700.0 (7760.0–17,871.0)	9927.2 (5461.3–17,414.2)	12,700.0 (8586.0–19,509.0)	10,645.3 (6052.1–17,870.9)	8700.0 (4663.1–16,907.7)	9133.8 (4747.7–16,546.6)	8057.6 (2997.1–15,146.6)	8300.0 (4003.0–17,441.0)	10,465.3 (6009.5–17,600.1)	10,645.3 (6052.1–17,870.9)	8700.0 (4663.1–16,907.7)
*p*-value		0.016		<0.001					0.068		0.034	
*p*-adj ^#^		0.176		0.006					0.990		0.544	
S3 *^,#^ (*N* = 491)	3176.6 (1787.1–5247.2)	4831.0 (2738.0–8480.0)	2976.7 (1689.8–4838.2)	4151.6 (2574.3–6179.4)	3323.1 (2068.6–5475.1)	2285.3 (1237.3–4795.7)	2806.8 (1404.4–5171.9)	2740.9 (1106.7–4583.0)	2920.1 (1423.5–5026.2)	3238.3 (1906.9–5255.8)	3323.1 (2068.6–5475.1)	2285.3 (1237.3–4795.7)
*p*-value		<0.001		<0.001					0.470		<0.001	
*p*-adj ^#^		<0.001		<0.001					0.990		0.008	
S4 * (*N* = 423)	1025.6 (614.8–1508.0)	1465.2 (1021.0–3559.1)	966.0 (583.6–1431.8)	1300.5 (948.7–1889.5)	1069.7 (686.7–1671.6)	819.4 (469.6–1411.5)	928.5 (579.7–1420.2)	816.6 (372.7–2156.6)	1033.0 (543.0–6997.0)	1025.6 (629.7–1494.1)	1069.7 (686.7–1671.6)	819.4 (469.6–1411.5)
*p*-value		<0.001		<0.001					0.990		<0.001	
*p*-adj ^#^		<0.001		0.003					0.990		0.002	
**Multiple Linear Regression Models**												
Response variable	Best model formula			Adjusted *R*^2^		*p*-value			Predictors’ significance in the model containing all predictors			
S1	log(S1) = 11.190−1.277 × log (Age) + 2.217 × COVID-19 before vaccination (0 = no, 1 = yes)			0.372		<0.001			COVID-19 before vaccination (*p* < 0.001), age (*p* < 0.001), patient contact (*p* = 0.136), sex (*p* = 0.166)			
S4	log(S4) = 0.145 × log (S1) + 0.628 × log (S2)			0.709		<0.001			S2 (*p* < 0.0001), S1 (*p* < 0.001), patient contact (*p* = 0.069), sex (*p* = 0.067), COVID-19 before vaccination (*p* = 0.224), age (*p* = 0.980)			

* Sera of participants obtained after COVID-19 onset were not included. ^#^ *p*-Values were adjusted for multiple comparisons with the Bonferroni method. S1 = sera obtained 3 weeks after the first vaccine dose. S2 = sera obtained 1 month after the second vaccine dose. S3 = sera obtained 3 months after the second vaccine dose. S4 = sera obtained 6 months after the second vaccine dose.

## Data Availability

The data presented in this study are available on request from the corresponding author. The data are not publicly available due to privacy and ethical restrictions.

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
