# Peer review of "Decline of Anti-SARS-CoV-2 IgG Antibody Levels 6 Months after Complete BNT162b2 Vaccination in Healthcare Workers to Levels Observed Following the First Vaccine Dose"

_vaccines, 2022, doi:10.3390/vaccines10020153_

Round 1

Reviewer 1 Report

In this manuscript, Dakovic Rode et al. analyze the anti-SARS-CoV-2 IgG antibody levels after the first dose and six months after complete Bnt162b2 vaccination in healthcare workers at the University Hospital for Infectious Diseases in Zagreb (Croatia). This longitudinal prospective study involved a cohort of 587 healthcare workers (age 18-66) in the country’s leading institution for the treatment of COVID-19 patients. Anti-SARS-CoV-2 IgG antibody levels were assessed from blood samples at four time points (three weeks after the first BNT162b2 vaccine dose, and then one, three and six months after the second vaccine dose from January 2021 to September 2021) using a quantitative chemiluminescent microparticle immunoassay for antibodies against the receptor-binding domain (RBD) of S1 subunit of the SARS-CoV-2 spike protein.

Overall, the paper is very well written and very informative and I do not have any major concern related to either the study design or the analysis of the data. However, I think that some small changes to the text are required before the manuscript can be accepted for publication in Vaccines:

  • Line 202-204: “In addition, HCWs in close contact with COVID-19 patients reached significantly higher antibody titers than other HCWs at all of the analyzed time points.” I find it difficult to understand this stratification parameter, when the whole analysis revolves around participants with and without a history of COVID-19. Are you suggesting that subjects who were in contact with COVID-19 patients may have developed a certain level of immunity even without becoming infected? I so, could you provide a tentative explanation for this observation? Are you referring to people who became infected but remained asymptomatic? Please explain.

  • Line 331-332: “…..while most HCWs with a history of COVID-19 showed a decline in IgG levels at the stated time point.” This statement does not appear to be supported by the data, as not statistical significance is associated with this observation. I’d rather rephrase this conclusion and point out that, unlike at the other three time points, where subjects with a history of COVID-19 always seem to show a higher level of antibodies in their blood (as stated in line 154-157), at one month after the second dose, the antibody levels in subjects with a history of COVID-19 are similar to those in naïve participants.

Author Response

Response to Reviewer 1 Comments:

Overall, the paper is very well written and very informative and I do not have any major concern related to either the study design or the analysis of the data. However, I think that some small changes to the text are required before the manuscript can be accepted for publication in Vaccines:

Point 1: Line 202-204: “In addition, HCWs in close contact with COVID-19 patients reached significantly higher antibody titers than other HCWs at all of the analysed time points.” I find it difficult to understand this stratification parameter, when the whole analysis revolves around participants with and without a history of COVID-19. Are you suggesting that subjects who were in contact with COVID-19 patients may have developed a certain level of immunity even without becoming infected? I so, could you provide a tentative explanation for this observation? Are you referring to people who became infected but remained asymptomatic? Please explain.

Response 1:

We are grateful for your opinion on our work. We agree with your observation. The sentence was based on the data from Table 1 for HCWs who were analysed according to workplace with close contact with COVID-19 patients. We hypothesized that during close contact with COVID-19 patients, despite that HCWs have used protective measures continuously, a small amount of virus could reach employees and induce an immune response without causing disease. At this point, it is difficult to differentiate passive immunization after a small amount of virus exposure and asymptomatic infection. Those HCWs who had a breakthrough SARS-CoV-2 infection after vaccination were identified by unexpected antibody findings, and we further individually checked their medical history, PCR findings, and analysed anti-NP SARS-CoV-2 antibodies. They were excluded from further analyses (3.1. Study Participants). We presented them in Fig. 2.

We have corrected the sentence to make it clearer and we hope you agree (lines 233-236):

In addition, we hypothesized that the transmission of a small concentration of SARS-CoV-2, which is insufficient to cause COVID-19, could lead to higher IgG titers in HCWs in prolonged close contact with COVD-19 patients. These HCWs reached significantly higher antibody titers than other HCWs at S1, S3 and S4.

Point 2: Line 331-332: “…..while most HCWs with a history of COVID-19 showed a decline in IgG levels at the stated time point.” This statement does not appear to be supported by the data, as not statistical significance is associated with this observation. I’d rather rephrase this conclusion and point out that, unlike at the other three time points, where subjects with a history of COVID-19 always seem to show a higher level of antibodies in their blood (as stated in line 154-157), at one month after the second dose, the antibody levels in subjects with a history of COVID-19 are similar to those in naïve participants.

Response 2:

Many thanks for this suggestion. We rephrased and adapted the conclusion (lines 363-371):

After completed two-dose BNT162b2 vaccination, HCWs showed a decrease in anti-SARS-CoV-2 IgG antibody levels as early as three months. History of COVID-19 showed a significant influence on antibody kinetics in vaccinated subjects. Unlike at the other three time points, where subjects with a history of COVID-19 always seem to show a higher level of antibodies, one month after the second dose, the antibody levels in subjects with a history of COVID-19 were similar to those in COVID-19 naïve participants. The COVID-19 naïve participants exhibited a multifold increase in antibody levels one month after the second dose which differs from kinetics in participants with a history of COVID-19. However, six months after vaccination, regardless of pre-vaccination history of COVID-19, all participants had anti-SARS-CoV-2 IgG levels similar to antibody levels reached by COVID-19 naïve participants three weeks after the first dose of vaccine. Our results suggest a strong decline in IgG levels six months after vaccination with two doses, which was strongly correlated with IgG levels three weeks after the first dose, as well as with IgG levels achieved one month after the second dose. However, further monitoring is needed to elucidate the relationship between IgG decay and infection protection.

Reviewer 2 Report

This manuscript presented the IgG antibody response after vaccination with the BNT162b2 SARS-CoV-2 mRNA vaccine in healthcare workers aimed to assess the post-vaccination anti-SARS-CoV-2 IgG dynamics. Overall, the manuscript has some scientific contribution in predicting the need for revaccination according to antibody levels after vaccination.

However, some minor revisions are needed before accept for publication in Vaccines Journal.  

  1. Time intervals between first and second doses should be described.
  2. The manuscript can be improved by expanding more detail in the analysis 1) antibody responses of pre-exposed vs non-exposed in different age groups and show them in chart with lines. 2) antibody responses of pre-exposed vs non-exposed in sex groups and show them in chart with lines.
  3. It is hard to understand the figure 2. Please remake it and describe more detail for understanding by readers.

Author Response

Response to Reviewer 2 Comments:

This manuscript presented the IgG antibody response after vaccination with the BNT162b2 SARS-CoV-2 mRNA vaccine in healthcare workers aimed to assess the post-vaccination anti-SARS-CoV-2 IgG dynamics. Overall, the manuscript has some scientific contribution in predicting the need for revaccination according to antibody levels after vaccination.

However, some minor revisions are needed before accept for publication in Vaccines Journal.  

Point 1: Time intervals between first and second doses should be described.

Response 1: Thank you for this note. We apologize to have missed that. We use a two-dose regimen of PfizerBioNTech COVID-19 vaccine BNT162b2 given three weeks apart (lines 73-74, 107).

Point 2: The manuscript can be improved by expanding more detail in the analysis 1) antibody responses of pre-exposed vs non-exposed in different age groups and show them in chart with lines. 2) antibody responses of pre-exposed vs non-exposed in sex groups and show them in chart with lines.

Response 2: We are grateful for your suggestions. Your suggestions are in line with our concerns and analysis, and we have already done an analysis of the antibody responses of previously exposed and unexposed subjects according to age groups and sex groups, but we did not consider the results important enough to show. We have attached the analyzed graphs with an explanation below. In Figure A, the analysis showed increased IgG titers in participants with a history of COVID-19 in samples S1, S3, and S4 in all age groups. The key difference between the age groups was the finding of significantly higher antibody titers in COVID-19-naïve participants in the 18-30 age group in S2 samples. The graph also shows the trend of increased differences between COVID-19-naïve and COVID-19 recovered participants in the older groups in samples S3 and S4. However, this finding is highly questionable due to the small number of older participants with a history of COVID-19. Given that these results are due to a moderate negative correlation between age and IgG titer in COVID-19-naïve participants, as mentioned in the manuscript, we believe these findings do not contain enough new information to be presented as a figure. According to the presented data, we have added the main findings of this analysis to the part with the results of the revised manuscript (lines 223-231 without including the charts).

We also analyzed differences in IgG levels between COVID-19 naïve and COVID-19 recovered participants stratified by sex and included Figure B below. This analysis showed an increase in IgG levels in participants with a history of COVID-19 in samples S1, S3, and S4, regardless of sex. Since sex was not significant in either bivariate or multivariate analyses, we believe that including this graph in the manuscript would not provide sufficient new information on differences between COVID-19-naïve and COVID-19 recovered participants, stratified by sex. This is further supported by the relatively small number of male participants with a history of COVID-19, which could hinder the possible conclusions of this analysis. We appreciate your suggestions and believe that you can accept our explanations and the reasons why we believe that the data in the charts, especially due to the small number in individual groups, were not important enough to improve the manuscript.

Figures A & B are attached in PDF File below.

Figure A. Comparison of IgG titers between COVID-19-naïve and COVID-19 recovered participants, stratified by age groups. Y-axis is logarithmically scaled for better resolution. S1: IgG titers three weeks after the first dose, S2: IgG titers one month after the second vaccine dose, S3: IgG titers three months after the second vaccine dose, S4: IgG titers six months after the second vaccine dose. ns: non-significant, *: p < 0.05, **: p < 0.01, ***: p < 0.001 (Mann-Whitney U test).

Figure B. Comparison of IgG titers between COVID-19-naive and COVID-19-recovered participants, stratified by sex. Y-axis is logarithmically scaled for better resolution. S1: IgG titers three weeks after the first dose, S2: IgG titers one month after the second vaccine dose, S3: IgG titers three months after the second vaccine dose, S4: IgG titers six months after the second vaccine dose. ns: non-significant, *: p < 0.05, **: p < 0.01, ***: p < 0.001 (Mann-Whitney U test).

Point 3: It is hard to understand the figure 2. Please remake it and describe more detail for understanding by readers.

Response 3: We apologize for not describing the chart clearly enough. The Figure 2 represents a histogram of ratios of IgG levels in each participant between samples taken A) one month after the second and three weeks after the first dose of vaccine, B) three and one month after the second dose of vaccine and C) six and three months after the second dose of vaccine We hope that we have remake it well enough and that it is now more understandable to readers. Below is a description of the figure inserted in the manuscript:

Figure 2. Histogram of ratios of SARS-CoV-2 IgG levels measured (A) one month after the second BNT162b2 vaccine dose (S2) and three weeks after the first vaccine dose (S1) (N = 524). B) three months after the second vaccine dose (S3) and one month after the second vaccine dose (N = 479). (C) six months after the second vaccine dose (S4) and three months after the second vaccine dose (N = 406). Participants are colored according to their history of COVID-19. The X-axis is logarithmically scaled for increased resolution. The dotted line marks IgG titer ratio of 1 which corresponds to no changes in IgG titer between the respective time points. Participants shown left of the dotted line exhibited a decrease in IgG titers between the respective time points. Participants shown right of the dotted line showed an increase in IgG titers between the respective time points.

Round 2

Reviewer 1 Report

The manuscript can now be accepted for publication.